# Structure and dynamics of the pyroglutamylated RF-amide peptide QRFP receptor GPR103

Aika Iwama[1], Ryoji Kise[2], Hiroaki Akasaka [1], Fumiya K. Sano [1], Hidetaka S. Oshima [1], Asuka Inoue [2,3] ✉, Wataru Shihoya [1] ✉ & Osamu Nureki [1] ✉

Pyroglutamylated RF-amide peptide (QRFP) is a peptide hormone with a C-terminal RF-amide motif. QRFP selectively activates a class A G-protein-coupled receptor (GPCR) GPR103 to exert various physiological functions such as energy metabolism and appetite regulation. Here, we report the cryo-electron microscopy structure of the QRFP26-GPR103-$G_q$ complex at 3.19 Å resolution. QRFP26 adopts an extended structure bearing no secondary structure, with its N-terminal and C-terminal sides recognized by extracellular and transmembrane domains of GPR103 respectively. This movement, reminiscent of class B1 GPCRs except for orientation and structure of the ligand, is critical for the high-affinity binding and receptor specificity of QRFP26. Mutagenesis experiments validate the functional importance of the binding mode of QRFP26 by GPR103. Structural comparisons with closely related receptors, including RY-amide peptide-recognizing GPCRs, revealed conserved and diversified peptide recognition mechanisms, providing profound insights into the biological significance of RF-amide peptides. Collectively, this study not only advances our understanding of GPCR-ligand interactions, but also paves the way for the development of novel therapeutics targeting metabolic and appetite disorders and emergency medical care.

Neuropeptides, a diverse array of signaling molecules, orchestrate a multitude of physiological processes in organisms. Among them, neuropeptides possessing the Arg-Phe-$NH_2$ (RF-amide) motif at their C-termini are designated as RF-amide peptides and have attracted considerable attention because of their pivotal roles in various biological functions[1–3]. The RF-amide family encompasses five distinct peptides: neuropeptide FF (NPFF), prolactin-releasing peptide (PrRP), kisspeptin (Kiss1), gonadotropin-inhibitory hormone (GnIH), and pyroglutamylated RF-amide peptide (QRFP). Each peptide interacts with specific class A G-protein-coupled receptors (GPCRs), initiating a cascade of intracellular events to modulate physiological responses. Similarly, the RY-amide peptides, characterized by the signature RY motif at their C-termini, present a similar mode of action, interacting with their respective receptors to regulate physiological processes[4]. This categorization based on the C-terminal residues highlights the specificity and diversity within these peptide families. The intricate interplay between these peptides and their receptors represents a complex network critical for maintaining homeostasis and responding to environmental changes.

[1]Department of Biological Sciences, Graduate School of Science, The University of Tokyo, Bunkyo, Tokyo 113-0033, Japan. [2]Graduate School of Pharmaceutical Sciences, Tohoku University, 6-3, Aoba, Aramaki, Aoba-ku, Sendai, Miyagi 980-8578, Japan. [3]Graduate School of Pharmaceutical Sciences, Kyoto University, 46-29 Yoshida-Shimo-Adachi-cho, Sakyo-ku, Kyoto 606-8501, Japan. ✉e-mail: iaska@tohoku.ac.jp; wtrshh9@gmail.com; nureki@bs.s.u-tokyo.ac.jp

QRFP is a 43-amino acid RF-amide peptide with a pyroglutamylated N-terminus (namely QRFP43)[5] and demonstrates specific activity for GPR103 with orexigenic activity[5–7]. GPR103 is a $G_{i/o}$- and $G_q$-coupled GPCR expressed in the adrenal gland and various regions of the brain such as the hypothalamus, which plays a crucial role in the regulation of energy metabolism and appetite control. QRFP and GPR103 are implicated in a variety of physiological functions ranging from the modulation of feeding behavior to the regulation of energy homeostasis[8], cardiovascular function, and bone formation[9]. This QRFP-GPR103 pair is remarkably conserved across various vertebrate species[10], highlighting its fundamental significance in biological systems. GPR103 is not only pivotal in maintaining physiological balance but also presents therapeutic targets for disorders related to metabolism and appetite dysregulation. For example, QRFP administration reportedly increased food intake and fat mass while reducing glucose-induced insulin release, and may also cause osteopenia and facilitate nociception[11]. Thus, GPR103 antagonists are expected to be useful in the prevention and treatment of various metabolic disorders such as bulimia, vasospasm, obesity, diabetes, endocrine disorders, hypercholesterolemia, hyperlipidemia, gout, and fatty liver. They may also be useful as therapeutic agents for cardiovascular and renal diseases, including atherosclerosis and heart failure. Several nonpeptidic GPR103 antagonists (e.g. pyrrolo[2,3-c]pyridine) have been discovered from peptidomimetics and high-throughput screening, and have demonstrated anorexic activity in mice.

Recent advancements in cryo-electron microscopy (cryo-EM) have unveiled numerous GPCR–G-protein complex structures, including those associated with C-terminally amidated peptides such as cholecystokinin, orexin, and RY-amide neuropeptide Y[12–14]. These analyses have shed light on the intricacies of ligand-receptor interactions and their activation mechanisms. Nevertheless, a notable gap remains in our understanding of the mechanisms distinguishing RF- and RY-amides. This also includes an understanding of the selective activation of GPR103 by QRFP and the fundamental principles dictating ligand specificity among related receptors. These gaps in structural data significantly hinder the strategic development of therapeutic agents targeting GPR103. Here, we show a cryo-EM structure of the QRFP-bound GPR103•$G_q$ complex, offering insights into its ligand recognition and selectivity, and dynamics.

## Results

### Overall structure

For the structural study, we used an N-terminally truncated form of QRFP, known as QRFP26 (Fig. 1a). QRFP26 is also found in vivo and possesses agonist activity comparable to that of QRFP43[7,8,15]. In this study, we focused on the structural analysis complexed with $G_q$. To facilitate expression and purification, we truncated the C-terminal residues after G366 of human GPR103. The truncated mutant activated $G_q$ to the same extent as the wild-type in the TGFα shedding assay[16] (Supplementary Fig. 1a–d and Supplementary Table 1). We used a mini-$G\alpha_q$ (mini-$G_{sqi}$), an engineered mini-$G\alpha_s$ protein whose N-terminal and C-terminal residues are replaced by $G\alpha_{i1}$ and $G\alpha_q$, respectively[17]. To efficiently purify the stable GPCR–G-protein complex, the receptor, and mini-$G_{sqi}$ were incorporated in a 'Fusion-G system' by combining two complex stabilization techniques[18,19] (Supplementary Fig. 2a, b). The modified receptor and G-protein were co-expressed in HEK293 cells and purified by FLAG affinity chromatography. After an incubation with Nb35 and scFv16 that bind to mini-$G_{sqi}$, the complex was purified by size-exclusion chromatography (Supplementary Fig. 2c, d). The structure of the purified complex was determined by single-particle cryo-EM analysis with an overall resolution of 3.19 Å (Supplementary Fig. 3, Supplementary Table 2, and "Methods"). As the extracellular portion of the receptor was poorly resolved, we performed receptor-focused refinement, yielding a density map with a nominal resolution of 3.29 Å, which was combined with the overall refined map.

The resulting composite map allowed us to precisely build the atomic model of all the components, including the receptor (residues 3–243 and 263–346), QRFP26 (residues 3–26), G-proteins, and antibodies (Fig. 1b, c).

The receptor consists of the canonical 7 transmembrane helices (TM) connected by three intracellular (ICL1–3) and three extracellular (ECL1–3) loops, the amphipathic helix 8 at the C-terminus (H8), and the N-terminal residues exposed on the extracellular side (Fig. 1d, e). ICL3 was disordered in the cryo-EM map. At the secondary-structure level, ICL2 and ECL3 contain short α-helices, and ECL2 forms a long β-sheet. The N-terminal residues extend above ECL2, constituting the extracellular domain (ECD) together with ECL2. An unambiguous density was observed from the interior of the transmembrane domain (TMD) to the ECD, allowing us to model residues 3–26 of QRFP26 (Fig. 1e). The residues 5–13 adopt an α-helix, consistent with the NMR analysis of QRFP alone[20]. The G-protein docks in the intracellular cavity of the receptor, forming similar interactions as in other G-protein complexes[13,17,19,21–25] involving conserved $D^{3.49}R^{3.50}Y^{3.51}$ (superscripts indicate Ballesteros–Weinstein numbers[26]) (modified to $E^{3.49}R^{3.50}H^{3.51}$ in GPR103) and $N^{7.49}P^{7.50}xxY^{7.53}$ motifs (Supplementary Fig. 4a–j).

### QRFP26 binding site in the transmembrane domain

QRFP26 binds to both the ECD and TMD with its C-terminal amide directed toward the TMD core (Fig. 1e). The C-terminal heptapeptide of QRFP26 (GGFSFRF) fits vertically within the TMD and creates an extensive interaction network with TMs 2–7 and ECL2 (Fig. 2a, b and Supplementary Fig. 5). This interaction is broadly divided into the residues 20–24 (GGFSF) and the RF-amide moiety (R25 and F26). In the former part, the peptide backbone of QRFP26 forms hydrogen bonds with $Q105^{2.64}$, $Q184^{4.64}$, and $Q318^{7.39}$. The $Q184^{4.64}A$ and $Q318^{7.39}A$ mutations reduced the potency of QRFP26 in the TGFα shedding assay, while the $Q105^{2.64}A$ mutation retained the potency (Fig. 2c, Supplementary Fig. 1a–c and Supplementary Table 1). Furthermore, F22 and F24 form stacking interactions with $W111^{ECL1}$ (Fig. 2b), and the $W111^{ECL1}A$ mutation abolished the response for QRFP26 (Fig. 2c, Supplementary Fig. 1a–c). $W111^{ECL1}$ stabilizes the upright structure of QRFP26 and plays an essential role in QRFP26 binding.

In comparison, RF-amide binds deep in the pocket (Fig. 2a), consistent with its significance in receptor binding reported in previous studies with mutant peptides[27,28]. The F26 side chain is present at the deepest position in the TMD, surrounded by bulky hydrophobic residues in TM2, TM3, and TM6 (Fig. 2d). The C-terminal amide forms hydrogen-bonding interactions with $T102^{2.61}$, $Q125^{3.32}$, and $Q318^{7.39}$, and these mutations reduced the QRFP26 potency by over 3-fold (Fig. 2c). R25 forms diverse interactions with receptors, including hydrogen bonds with $T215^{5.39}$, electrostatic interactions with $E203^{ECL2}$ and $E297^{6.59}$, and stacking interactions with $W205^{ECL2}$ and $Y214^{5.38}$ (Fig. 2e). While the $T215^{5.39}A$ mutation minimally affected the potency, the $E203^{ECL2}A$ and $E297^{6.59}A$ mutations reduced the potency by 3- and 10-fold, respectively, and the $E203^{ECL2}A/E297^{6.59}A$ double mutation abolished the potency (Fig. 2c, Supplementary Fig. 1a–c, Supplementary Table 1). These results indicate that electrostatic interaction with $E203^{ECL2}A$ and $E297^{6.59}A$ plays an important role in QRFP26 binding. $E203^{ECL2}$ and $E297^{6.59}$ make the ligand-binding pocket of the TMD negatively charged (Fig. 2f). Thus, the C-terminal amidation functions not only in interactions with the receptor but also in enhancing the charge complementarity with the pocket, by neutralizing the negative charge of the C-terminal carboxylate. Moreover, the $W205^{ECL2}A$ and $Y214^{5.38}A$ mutations dramatically reduced the potency (Fig. 2c, Supplementary Fig. 1b, c, Supplementary Table 1). These aromatic residues would also be involved in pocket formation through the stacking interactions between them (Fig. 2e), suggesting their importance. Overall, our structural and mutational studies demonstrate the structure-activity relationship of QRFP26 binding in the TMD.

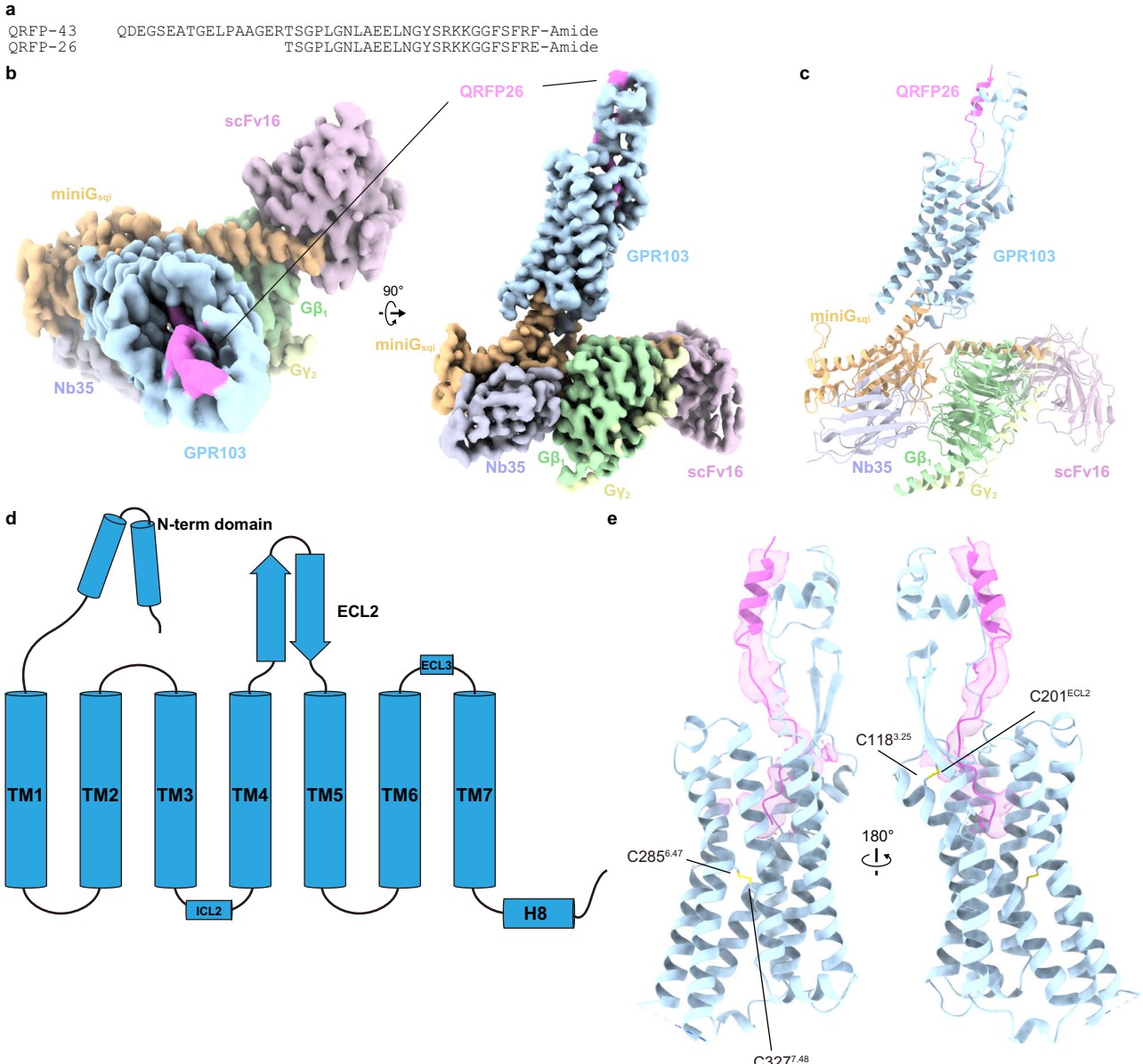

**Fig. 1 | Overall structure of the GPR103–mini-G$_{sqi}$β$_1$γ$_2$–scFv16–Nb35 complex.**
**a** Amino acid sequences of QRFP43 and QRFP26. **b** Cryo-EM density map of the GPR103–mini-G$_{sqi}$β$_1$γ$_2$–scFv16–Nb35 individually colored. **c** Refined structures of the complex are shown as a ribbon representation. **d** Diagram of GPR103. N-terminal forms a helix-loop-helix motif. ECL2 forms a long β-sheet. **e** Ribbon representation of the QRFP26 and GPR103. Density focused on QRFP (pink). Two disulfide bonds are represented by stick models. The one is the highly conserved disulfide bond between C118$^{3.25}$ and C201$^{ECL2}$, and the other is atypical disulfide bond between C285$^{6.47}$ and C327$^{7.48}$. The C118$^{3.25}$A and C201$^{ECL2}$A mutations abolished the QRFP26 potency (Supplementary Fig. 1a–d). By contrast, the C285$^{6.47}$A and C327$^{7.48}$A mutations did not alter the potency, indicating their lesser importance for receptor function.

Previous studies with mutant peptides have shown that the C-terminal heptapeptide is sufficient to activate GPR103[27,28], despite its reduced affinity. The heptapeptide is conserved from fish to mammals, except for S23 (Supplementary Fig. 6a), consistent with the fact that the S23 side chain poorly interacts with the receptor. The receptor residues interacting with the heptapeptide are highly conserved among the homologs (Supplementary Fig. 6b, c). Accordingly, this observed heptapeptide-TMD interaction plays a key role in evolutionarily conserved GPR103 activation.

### Architecture of the N-terminal region
We observed an unambiguous density above ECL2, despite the suboptimal local resolution ranging from 4 to 6 Å (Supplementary Fig. 3). The density aligned with the N-terminal structure predicted by AlphaFold[29–31] and allowed us to discuss the potential interactions between the ECD and QRFP26. The N-terminal residues 2–40 of GPR103 extend from TM1 to above ECL2, partially covering the ligand-binding pocket in the TMD. The residues 2–18 form a helix-loop-helix (HLH) above ECL2 (Fig. 3a). QRFP26 extends vertically from G20, interacting predominantly with the N-terminal region. The peptide backbone of R17 and K18 form chain-to-chain hydrogen-bonding interactions with L34$^{N-term}$ and V35$^{N-term}$ (Fig. 3b), while the α-helix is recognized by HLH (Fig. 3c). Specifically, the HLH structure is stabilized by stacking between centrally oriented F11$^{N-term}$, F25$^{N-term}$, and Y29$^{N-term}$. These aromatic residues, together with surrounding hydrophobic residues, sandwich the α-helix of QRFP and form an extensive hydrophobic interaction. Moreover, the position of HLH is stabilized by hydrophobic interactions with ECL2 among

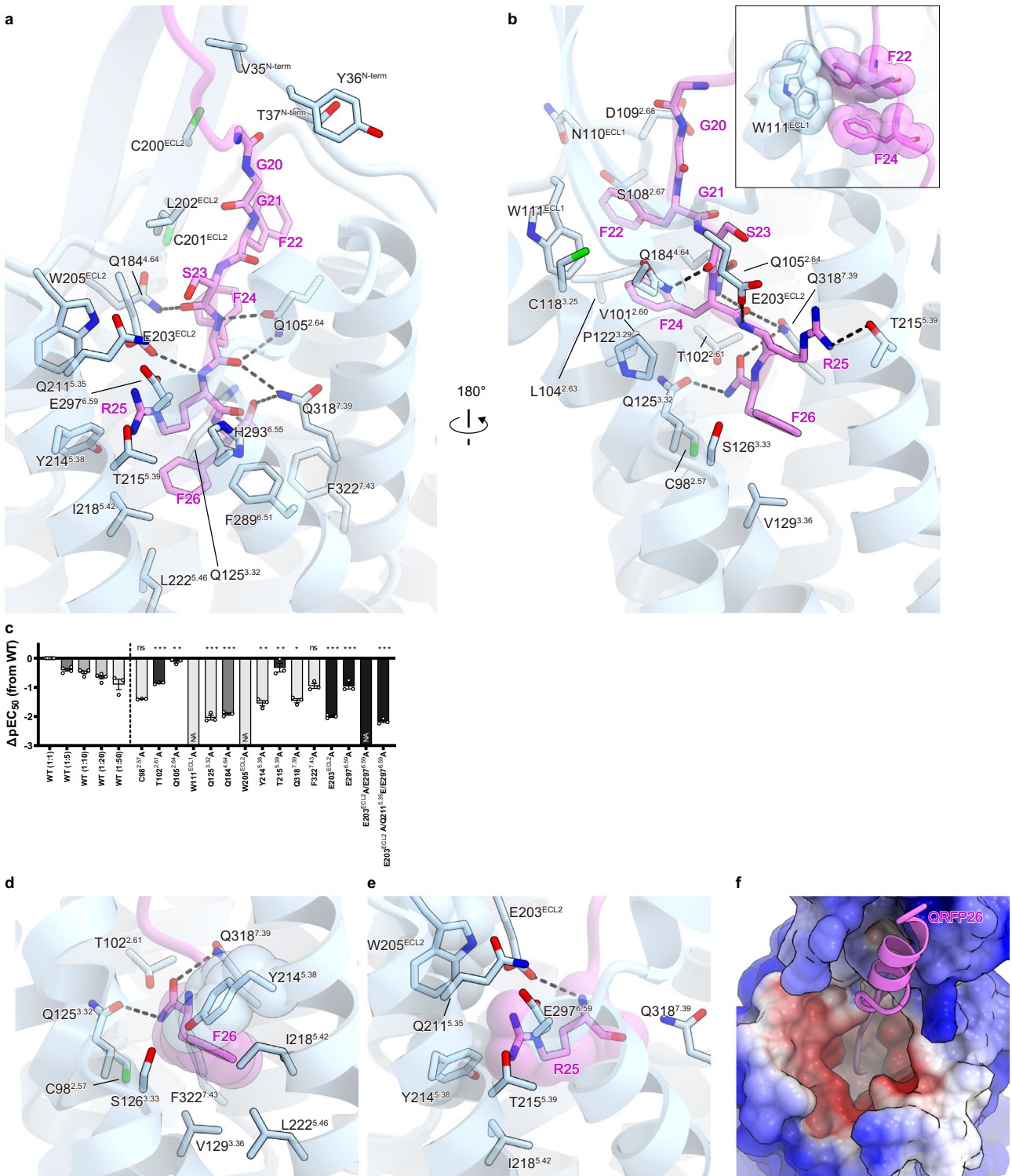

**Fig. 2 | QRFP26 binding site in the transmembrane region. a, b** Binding pocket for QRFP26 in the TMD. Residues involved in the QRFP-GPR103 interaction within 4.5 Å are shown as pink and blue sticks, respectively. Black dashed lines indicate hydrogen bonds. **c** Effects of mutations in the ligand-binding pocket of GPR103. QRFP26-induced activation of GPR103 was analyzed by the TGFα shedding assay. From the concentration–response curves (Supplementary Fig. 1b), ΔpEC₅₀ values relative to the wild-type were calculated. Colors in the mutant bars indicate an expression level matching that of titrated wild-type. NA, parameter not available because of lack of the ligand response. WT, wild-type. Statistical analyses were performed using the ordinary one-way ANOVA followed by Dunnett tests with the expression-matched (colored) wild-type response. ns, $p > 0.05$; *$p < 0.05$; **$p < 0.01$; ***$p < 0.001$. Data are presented as mean values ± SEM from at least three independent experiments performed in triplicate ($n = 5$ for the wild-type and $n = 3$ for the mutants). Source data are provided as a Source Data file. **d, e** Close-up views of F26 (**d**) and R25 (**e**) of QRFP26. **f** Surface representation of GPR103 viewed from the extracellular side. Positive and negative charges of the receptor are colored in blue and red, respectively.

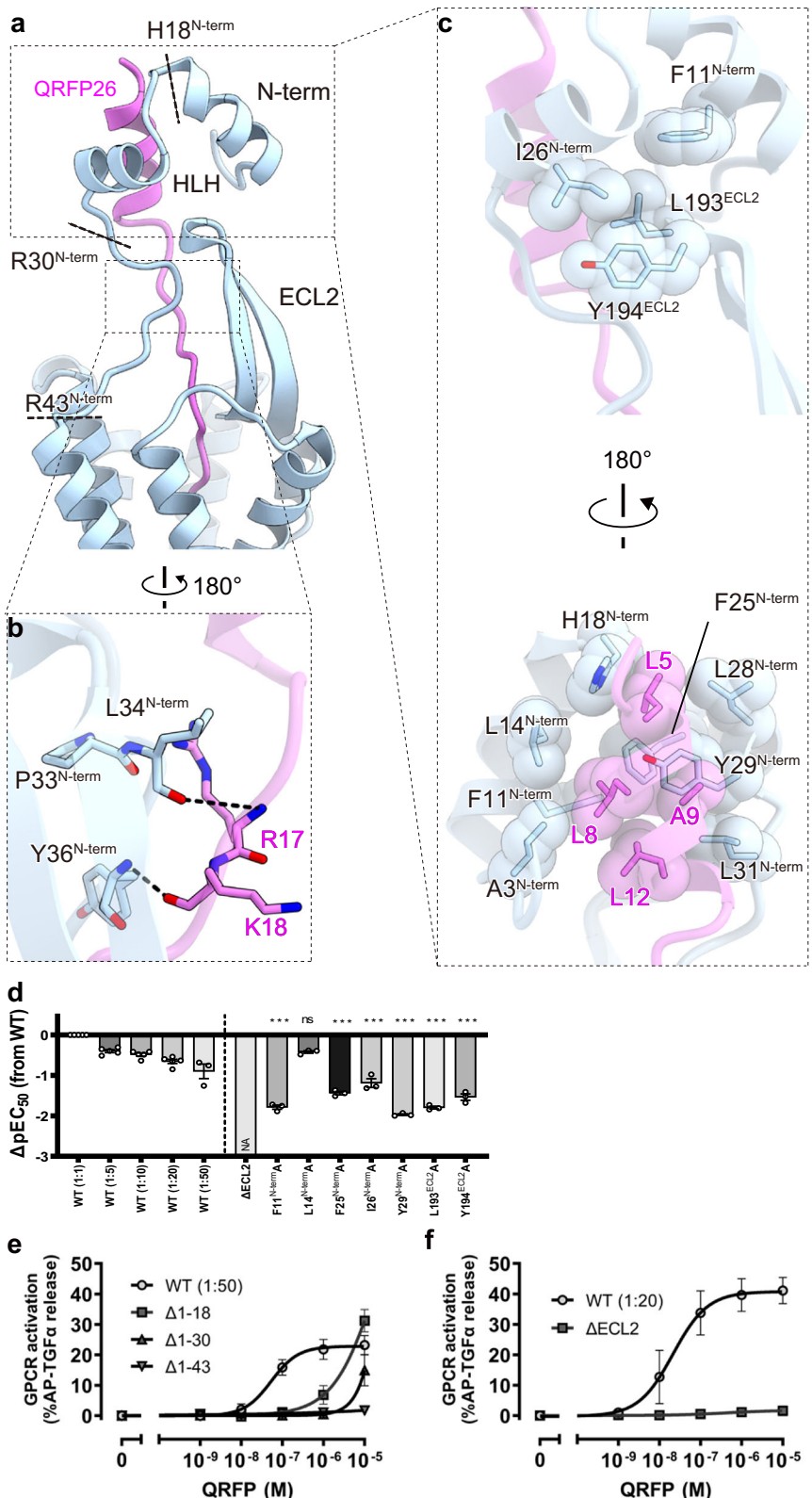

**Fig. 3 | Architecture of the extracellular region. a** Ribbon representation of GPR103 focused on the extracellular side. **b** Hydrogen-bonding interactions of R17-P33[N-term] and K18-Y36[N-term] are indicated by black dashed lines. **c** Hydrophobic interaction of the N-terminal HLH with ECL2 and QRFP26. **d**–**f** Effects of mutations in the ECD of GPR103. QRFP26-induced activation of GPR103 was analyzed by the TGFα shedding assay. From the concentration–response curves (Supplementary Fig. 1a), $\Delta pEC_{50}$ values relative to the wild-type were calculated (**d**). The concentration–response curves of the N-terminus-truncated mutants and the ECL2 deletion mutant were shown in **e** and **f**, respectively. Data for the wild-type

response was obtained from the same experiment as Fig. 2c. Colors in the mutant bars indicate an expression level matching that of titrated wild type. NA, parameter not available because of lack of the ligand response. WT, wild-type. Statistical analyses were performed using the ordinary one-way ANOVA followed by Dunnett tests with the expression-matched (colored) wild-type response. ns, $p > 0.05$; $*p < 0.05$; $**p < 0.01$; $***p < 0.001$. Data are presented as mean values ± SEM from at least three independent experiments performed in triplicate ($n = 5$ for the wild-type and $n = 3$ for the mutants). Source data are provided as a Source Data file.

F11[N-term], I26[N-term], L193[ECL2], and Y194[ECL2] (Fig. 3c). The alanine mutations of these aromatic residues reduced the QRFP26 potency by about 10-fold (Fig. 3d, Supplementary Fig. 1a–c, Supplementary Table 1). Overall, our structure and mutational study indicate that N-terminal HLH plays a critical role in the binding of QRPF26.

To further examine the functional role of the ECD, we made truncated mutants of the N-terminal region and ECL2 (Fig. 3a). Δ1–18 or Δ1–30 is the mutant in which the N-terminal HLH is partially and completely truncated, respectively. Δ1–43 or ΔECL2 is the mutant in which the N-terminal region or ECL2 is completely truncated, respectively. Although the expression levels of these mutants were greatly reduced (Supplementary Fig. 1b), we could analyze the effects of the mutations by comparing them with the wild-type, whose expression levels were reduced to the same extent. Δ1–43 and ΔECL2 completely lost the QRFP26 potency in the TGFα shedding assay (Fig. 3e, f). Such truncation would completely break the ECD architecture and further affect the shape of the ligand-binding pocket in the TMD, suggesting that the ECD plays an indispensable role in QRFP binding. By contrast, we measured the QRFP26-dependent activity of Δ1–18 and Δ1–30 at concentrations of $10^{-6}$ to $10^{-5}$ M (Fig. 3e), although their potencies were greatly reduced to the point that their $EC_{50}$ values could not be calculated. These data suggest that the HLH fractions as an affinity trap, achieving high compatibility through interactions with the N-terminal residues of QRFP26. Notably, Δ1–18 was more active than Δ1–30 (Fig. 3e), suggesting the remaining function as the affinity trap in Δ1–18 and further validating our cryo-EM structure. This N-terminal HLH motif is evolutionarily conserved among GPR103 homologs from zebrafish to humans (Supplementary Fig. 6c), and our homology searches have failed to identify any similar sequences in other proteins, indicating an unusual feature of GPR103.

Next, we performed a 3D flexible refinement implemented in cryoSPARCv4.4.0[32] (Supplementary Fig. 7a–f) and uncovered a significant conformational change of the ECD (Supplementary Fig. 7g, h and Supplementary Movie 1). We then built the models of this alteration on the two most significantly changed maps among the output (Fig. 4a. b and Supplementary Table 2). The results revealed the upright and tilted states of the ECD. A structural comparison of the two states elucidated the dynamic movement of the N-terminal HLH by 4.6 Å (Fig. 4b). Moreover, QRFP26, the ECD, and the extracellular half of the TMD oscillate like a pendulum with the C-terminus of QRFP26 as the base point (Supplementary Movie 1). In the original map, the ECD of the refined structure is positioned in between the upright and tilted states, whereas the transmembrane helices are more closely aligned with the upright, implying the fundamental stability of the upright state.

Similar fluctuating action is observed in the parathyroid hormone receptor 1 (PTH1R), a class B1 GPCRs that feature an ECD with a ~150 amino acid 'hotdog-like' domain. The ECD acts as an affinity trap by encasing the agonist peptide, and cryo-EM structural analysis elucidated the structural polymorphism in its ECD[33–35] (Fig. 4c, d). Despite significant differences in sequence homology and length, a functional analogy exists between the ECDs of GPR103 and PTH1R. However, the extracellular half of the GPR103-TMD undergoes a structural change following the ECD (Fig. 4b), while the PTH1R-TMD of PTH1R does not (Fig. 4d). Consequently, the structure and dynamics of the ECD and QRFP26 observed in this study are characteristic to GPR103, thus playing a critical role in the GPR103 selectivity of QRFP26.

### Conservation among the RF-amide receptors

The RF-amide moiety is the only conserved part of the RF-amide peptide, while the length and sequence of their N-terminal portions vary widely (Supplementary Fig. 8a). Thus, ligand interactions in the extracellular region are likely to be different for individual RF-amide receptors. Indeed, amino acid sequence alignment indicates that only

GPR103 has an N-terminal HLH (Supplementary Fig. 8b). To examine the conservation of the RF-amide recognition, we compared the residues interacting with the RF-amide with the corresponding RF-amide receptors; PrRPR (GPR10), KISS1R (GPR54), NPFFR1 (GPR147), and NPFFR2 (GPR74) (Fig. 5a). The recognition of the F26 side chain is less stringent but is shared by a bulky hydrophobic amino acid. By contrast, T102[2.61] and Q125[3.32], which recognize the C-terminal amide of QRFP26, are completely conserved. Although Q318[7.39] is replaced by histidine in all the other RF-amide receptors, it would form hydrogen bonds with the oxygen atom of the C-terminal amide of QRFP26. These considerations suggest that amide recognition by hydrogen-bonding interactions via these three residues is a common mechanism in RF-amide receptors. The two negative charges near R25 are also conserved in RF-amide receptors, although E297[6.59] is replaced by alanine in KISS1R. Instead, Q211[5.35] is replaced by E201[5.35] in KISS1R, suggesting the conserved recognition of R25 by the two negative charges. The triple mutants of E203[ECL2]A, E297[6.59]A, and Q211[5.35]E increased the QRFP26 potency compared with the E203[ECL2]A/E297[6.59]A double mutant (Fig. 2c, Supplementary Fig. 1a–d, and Supplementary Table 1). Thus, the RF-amide recognition mechanism observed in QRFP26–GPR103 is highly conserved in RF-amide receptors.

### Structural comparison with related peptide receptors

GPR103 exhibits a sequence homology with cholecystokinin receptors (CCKRs), orexin receptors (OXRs), and RY-amide neuropeptide Y receptors (YRs), which commonly recognize peptides with amidated C-termini (Supplementary Fig. 8a). To elucidate the characteristics of the structure and peptide recognition mechanisms of GPR103, we compared the endogenous ligand-bound structures of Y1R, CCK1R, and OX2R[12–14] (Fig. 5b). These receptors have a long ECL2 in common, with diversity in length and angle depending on the type of ligand. While the short peptide ligands cholecystokinin octapeptide (CCK-8) and orexin-B (OxB) are buried in the TMD, QRPF26, and neuropeptide Y (NPY) have an α-helix at the N-terminus. The α-helix of NPY is recognized by ECL2 that of QRFP26 is by N-terminal HLH, further highlighting the distinctiveness of the N-terminal structure for peptide recognition. Despite the differences on the extracellular side, the C-termini of the agonist peptides as well as the intracellular sides of the receptors, superimposed well, suggesting an evolutionary linkage among these receptors.

We compared the interactions of the C-terminal residues of the ligand in GPR103 and Y1R (Fig. 5c, d), representing the RF- and RY-amide receptors, respectively. In both instances, the C-terminal arginine of the ligand forms electrostatic interactions with two negatively charged residues. The carbonyl oxygen of the C-terminal amide hydrogen bonds with Q[3.32] and Q/H[7.39]. These four residues are highly conserved in the RF-amide and RY-amide receptors, indicating a shared recognition mechanism for the C-termini of the RX-amide peptide ligands (Fig. 5e). It should be noted that C[2.57] is completely conserved and forms a hydrogen bond with the C-terminal nitrogen of NPY in Y1R, while the hydrogen bond is doubtful in GPR103 due to bond angle issue. However, the C98[2.57]A mutation reduced the QRFP potency by 3-fold, suggesting the potential involvement in QRFP26 binding (Fig. 5e).

Next, we focused on the difference between the recognition of the RY-amide and RF-amide. In Y1R, Q219[5.46] forms a hydrogen bond with the hydroxyl group of the C-terminal Y36 of NPY[14,36] (Fig. 5c), which is highly conserved in the NPY receptors (Fig. 5a). Exceptionally, Q219[5.46] is replaced by L227[5.46] in Y2R (Fig. 5a), but its cryo-EM structure revealed that S223[5.42], situated one turn above, alternatively forms a hydrogen bond with Y36[14,37]. Contrarily, in the QRFP26–GPR103 complex, the hydrogen bond between Y36 and Q219[5.46] is replaced by a hydrophobic interaction between F26 and L222[5.46]. L[5.46] is highly conserved in the RF-amide receptors (Fig. 5a). Although L[5.46] is replaced by the polar residue T[5.46] in PrRPR, it is not

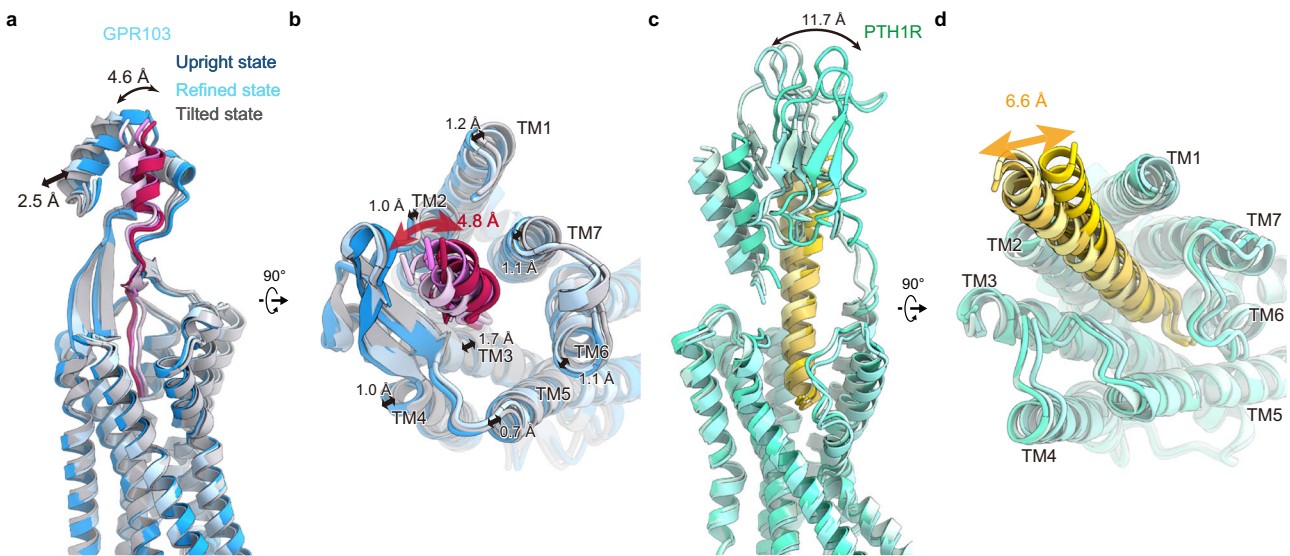

**Fig. 4 | Structural polymorphism of the ECD. a, b** Superposition of the GPR103 structures in the tilted, upright, and refined states. **c, d** Superimposition of the PTH1R structures in class 1–3 (PDB 7VVM (PTH1R-G$_s$ structure), 7VVL (PTH1R-G$_s$ structure), and 7VVK (PTH1R-G$_s$ structure)).

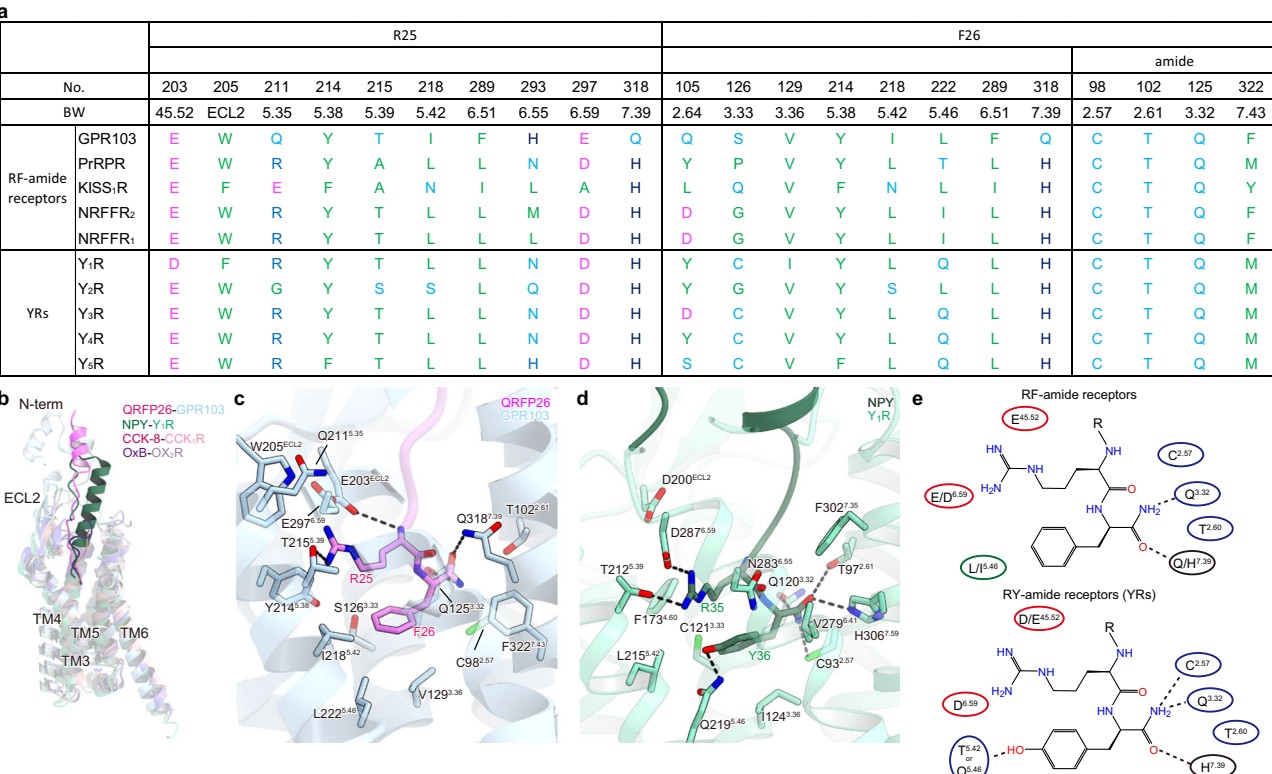

**Fig. 5 | Comparison of other related amide peptide receptors. a** Comparison of residues interacting with the RX-amide moiety in the RF- and RY-amide receptors. **b** Superimposition of GPR103 (blue), Y$_1$R (PDB 7X9A (NPY–Y$_1$R–G$_i$ structure), green), CCK$_1$R (PDB 7MBY (CCK-8–CCK$_1$R–mini-G$_{sqi}$ structure), magenta), and OX$_2$R (PDB 7L1U (OxB–OX$_2$R–mini-G$_{sqi}$ structure), purple) structures in complex with QRFP26, NPY, CCK-8, and OxB, peptides, respectively. NPY, neuropeptide Y; CCK-8, cholecystokinin octapeptide; OxB, orexin-B. **c, d** Structural comparison of the binding mode of the RF-amide moiety in GPR103 (**c**) and the RY-amide moiety in Y$_1$R (**d**). The residues involved in ligand-receptor interactions are represented by stick models. Black dashed lines indicate hydrogen bonds. **e** Schematic representations of RX-amide recognition conserved in RF- or RY-amide receptors. The most conserved residues are indicated based on the structures of GPR103 and Y$_1$R.

expected to interact with the C-terminal phenylalanine, due to its shorter side chain. The presence or absence of a residue capable of hydrogen bonding with the hydroxyl group of tyrosine at the C-terminus may distinguish the RF- and RY-amide receptors (Fig. 5e).

## Discussion

Our study has revealed the binding mode of the C-terminal QRFP heptapeptide to the TMD, which is sufficient for the activation of GPR103. The conserved recognition mechanism of the C-terminal RF-amide in various RF-amide receptors suggests broader biological

significance. Comparisons with other evolutionarily close peptide receptors demonstrated the commonality and diversity in the recognition mechanisms of the C-terminal two residues of peptide ligands. In particular, the presence or absence of residues that can form hydrogen bonds with the C-terminal side chain determines whether they function as RF- or RY-amide receptors. We identified the N-terminal HLH structure of GPR103, which captures the N-terminal side of QRFP and is quite unusual compared to class A GPCRs. We observed a pendulum-like motion of the ECD, including the N-terminus and the entire pocket, reminiscent of class B1 GPCRs. It should be noted that the orientation and secondary structure of the peptide ligands totally differ since the C-terminus of the α-helical ligand binds the extracellular domain in class B1 GPCRs. The structure and dynamics of these extracellular regions are important for promoting the specific, high-affinity binding of QRFP to GPR103. The distinctive structure and function of GPR103 determined in this study will be useful in the design of potential therapeutics targeting GPR103 for energy metabolism and appetite regulation.

Although the current active conformation was modeled based on the AlphaFold2 predicted structure[31], there are considerable differences between them. Notably, the intracellular side of TM6 is in a closed configuration (Supplementary Fig. 9a, b), and R143$^{3.50}$ of the D$^{3.49}$R$^{3.50}$Y$^{3.51}$ motif (modified to E$^{3.49}$R$^{3.50}$H$^{3.51}$ in GPR103) forms a salt bridge with E142$^{3.49}$ and is not oriented toward the intracellular face (Supplementary Fig. 9c), indicating that the predicted conformation represents the inactive state. This observation allowed us to discuss the mechanism of receptor activation upon QRFP26 binding, by comparing the cryo-EM structure with the predicted structure. Within the ligand-binding pocket in the TMD, conformational changes were observed in TM5 and TM6 (Supplementary Fig. 9d, e). The hydrogen-bonding interaction between R25 and T215$^{5.39}$ displaces TM5 inward by 1.4 Å. Conversely, F26 in QRFP induces the steric outward movement of F289$^{6.51}$, resulting in the outward displacement of TM6 by 1.6 Å. This movement of F26 drives W286$^{6.48}$ downward, thereby exerting a downward force on F282$^{6.44}$ in the underlying P$^{5.50}$I$^{3.40}$F$^{6.44}$ motif. This action is an important component of the P$^{5.50}$I$^{3.40}$F$^{6.44}$ motif reorganization frequently observed in class A GPCRs[21] and leads to the opening of the intracellular machinery of TM6. The mutations of F282$^{6.44}$A, W286$^{6.48}$A, and F289$^{6.51}$A almost abolished the potency of QRFP26 (Supplementary Fig. 1b, c, 9f, and Supplementary Table 1), supporting our proposed importance for receptor activation. Such receptor activation, characterized by conformational changes in W286$^{6.48}$, is a widespread phenomenon in class A GPCRs (e.g., human endothelin ET$_B$ receptor)[17–19,38–42].

In summary, these insights offer a comprehensive understanding of the mechanism of GPR103 activation upon QRFP binding. In the apo state, the N-terminal structure is hypothesized to be more labile than observed (Fig. 6a), due to its instability with merely two helices and a lower AlphaFold predictive score. QRFP may initially interact with either the ECD or TMD (Fig. 6b). Eventually, QRFP establishes stable binding to both domains, presumably maintaining the interaction of the C-terminus with the receptor while allowing for some fluctuation in the ECD and ligand-binding pocket in the TMD (Fig. 6c). This binding would reorganize the aromatic residue cluster in TM6, leading to the intracellular opening for receptor activation.

GPCRs except for class A commonly possess N-terminal domains with various lengths and functional roles (e.g. PTH1R[33]) (Fig. 7a), depending on the class. Although most class A GPCRs simply consist of 7TM, recent cryo-EM structures have visualized functional N-terminal regions, A typical example, in thyroid-stimulating hormone receptor (TSHR[43]), a large extracellular domain with multiple leucine-rich repeats captures the ligand protein (Fig. 7b). In chemokine receptor CXCR2[44] and C5a receptor (C5aR[45]), the N-terminus is elongated to the extracellular side and participates in the interaction with ligand protein (Fig. 7c, d).

Although not as large as the LRR or the hotdog-line domain, the N-terminal HLH of GPR103, interacts with QRFP26 together with its elongated N-terminal region (Fig. 7e). In this respect, GPR103 is intermediate between GPCRs with large extracellular domains and others (CXCR2 and C5aR).

Furthermore, structural comparisons revealed unexpected homology in peptide ligand recognition between C5aR and GPR103 (Fig. 7d, e). As QRFP26, C5a has its C-terminus buried in the TMD, from which the α-helix extends vertically. C5a has several helices in its additional region, some of which closely resemble the N-terminal HLH of GPR103 and interact similarly with ECL2. This similarity is probably coincidental since C5aR and GPR103 are not strikingly sequence homologous compared to other GPCRs. For stable receptor-ligand interactions at the extracellular side, the C5a–C5aR pair has a larger ligand. By contrast, the QRFP26–GPR103 pair has the HLH at the N-terminus of the receptor to capture the ligand.

In the current study, we have not only elucidated the architecture of the receptor but also shed light on the structure of QRFP itself, potentially paving the way for the modulation of QRFP-neurons (Q-neurons)[46]. QRFP is a biomarker for neurons and is capable of inducing an extended state of hypothermia and reduced metabolism, akin to hibernation. The QRFP-induced hibernation-like state (QIH) provides an unusual model to study metabolic suppression for therapeutic purposes, as well as artificial hibernation. QIH through pharmacogenetics[47] or optogenetics[48] is achievable in murine models, but the prospect of non-genetically triggering these neurons might represent a groundbreaking stride in the realm of synthetic hibernation research. Nevertheless, Q-neurons constitute a remarkably limited subset, and currently, there are no established methodologies for their selective pharmacological activation. Among the myriad strategies proposed, one promising avenue is the engineering of synthetic low-molecular-weight binders targeting QRFP, with their integration into novel drug delivery systems. The field of protein design is undergoing a rapid metamorphosis, fueled by recent innovations in structural prediction via AlphaFold[31] and the utilization of generative AI[46]. This is exemplified by the development of soluble GPCR variants and binders based on GPCR structures. Harnessing our structural insights to conceive binders that specifically target QRFP could facilitate studies of Q-neurons. Although this idea is still in its nascent stage, it harbors substantial promise for future applications, e.g. in emergency medical care.

## Methods

### Expression and purification of scFv16 and Nb35
The His$_8$-tagged scFv16 was expressed and secreted by Sf9 insect cells, as previously reported[49]. The Sf9 cells were removed by centrifugation at $5000 \times g$ for 10 min, and the secreta-containing supernatant was combined with 5 mM CaCl$_2$, 1 mM NiCl$_2$, 20 mM HEPES (pH 8.0), and 150 mM NaCl. The supernatant was mixed with Ni Superflow resin (GE Healthcare Life Sciences) and stirred for 1 h at 4 °C. The collected resin was washed with buffer containing 20 mM Tris (pH 8.0), 500 mM NaCl, and 20 mM imidazole, and further washed with 10 column volumes of buffer containing 20 mM HEPES (pH 8.0), 500 mM NaCl, and 20 mM imidazole. Next, the protein was eluted with 20 mM Tris (pH 8.0), 500 mM NaCl, and 400 mM imidazole. The eluted fraction was concentrated and loaded onto a Superdex200 10/300 Increase size-exclusion column, equilibrated in a buffer containing 20 mM Tris (pH 8.0) and 150 mM NaCl. Peak fractions were pooled, concentrated to 5 mg/mL using a centrifugal filter device (Millipore 10 kDa MW cutoff), and frozen in liquid nitrogen.

Nb35 was prepared as previously reported[50,51]. In brief, Nb35 was expressed in the periplasm of *E. coli*. The harvested cells were disrupted by sonication. Nb35 was purified by nickel affinity chromatography, followed by gel-filtration chromatography, and frozen in liquid nitrogen.

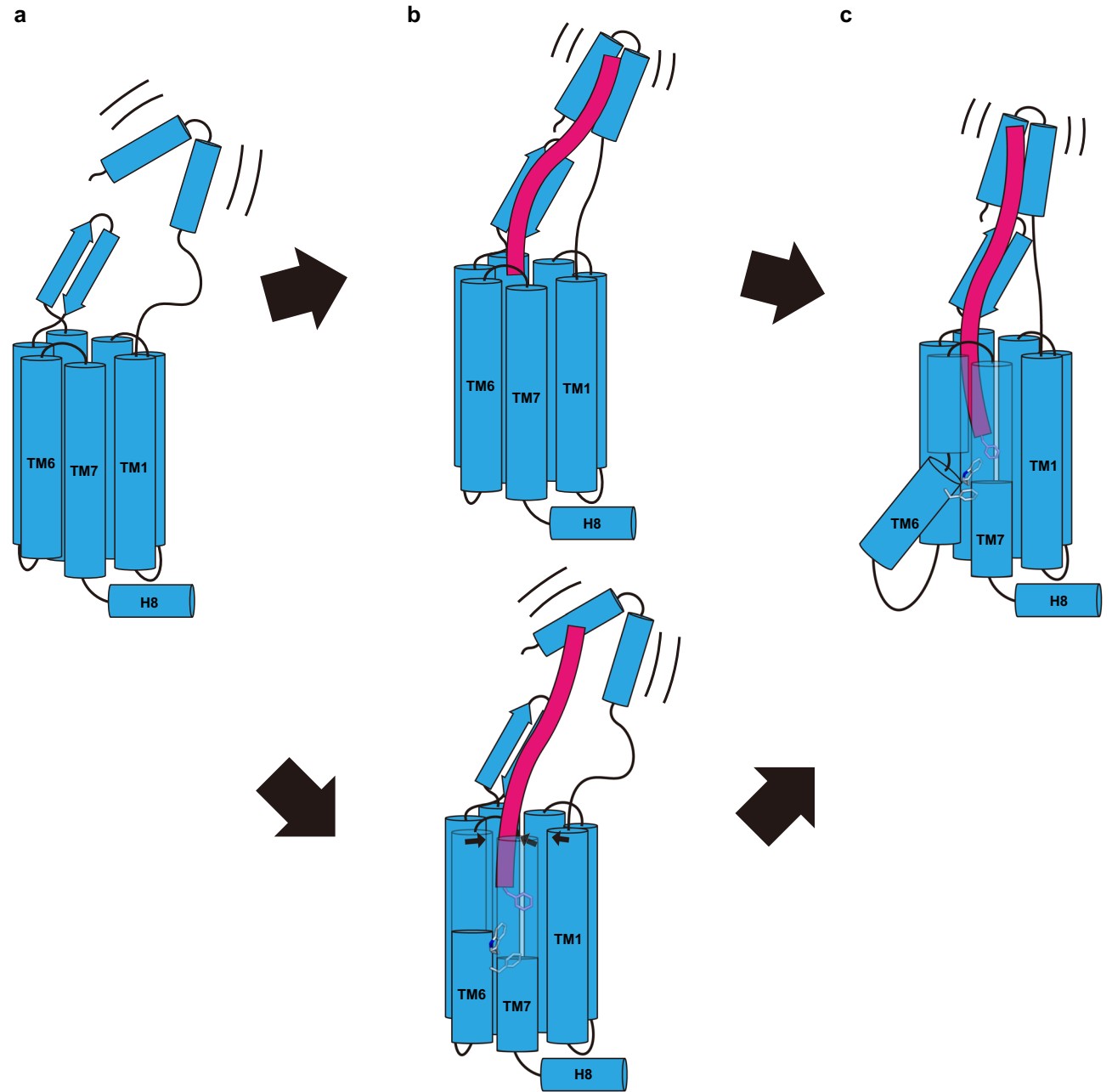

**Fig. 6 | Model of receptor activation. a–c** Schematic representations of GPR103 activation model upon QRFP binding. From the apo state (**a**), the peptide binds (**b**) and changes to the active state (**c**).

## Constructs for expression of GPR103 and $G_q$

The human GPR103 gene (UniProtKB, Q96P65) was subcloned into a modified pEG bacmamvector, with an N-terminal haemagglutinin signal peptide followed by the FLAG epitope (DYKDDDDK) and the LgBiT fused to its C-terminus followed by a 3C protease site and EGFP-His$_8$ tag. A 15 amino sequence of GGSGGGGGSGGSSSGG was inserted into both the N-terminal and C-terminal sides of LgBiT. Rat Gβ$_1$ and bovine Gγ$_2$ were subcloned into pEG vectors respectively. In detail, rat Gβ$_1$ was cloned with a C-terminal HiBiT connected with a 15 amino sequence of GGSGGGGGSGGSSSGG. Moreover, mini-G$_{sqi}$ was subcloned into the C-terminus of the bovine Gγ$_2$ with a nine amino sequence GSAGSAGSA linker. The two resulting pEG vectors can express the mini-G$_{sqi}$ trimer.

For the TGFα shedding assay and the flow cytometry analysis, we used human full-length QRFPR-encoding plasmids. Specifically, GPR103 was N-terminally modified with an HA signal sequence followed by a FLAG epitope and then a flexible linker (MKTIIALSYIFCLVFA**DYKDDDDK**GGSGGGGGSGGSSSGGG; the FLAG epitope is underlined) and inserted into the pCAGGS vector. Plasmid encoding GPR103 mutants were generated using Quickchange Site-Directed Mutagensis kit (Agilent).

## TGFα shedding assay

The TGFα shedding assay was performed as described previously with minor modifications[16]. Plasmid transfection was performed in a 6-well plate with a mixture of 500 ng (per well in a 6-well plate) AP-TGFα-encoding plasmid and 200 ng GPR103-encoding plasmid. After 1-day culture, the transfected cells were harvested by trypsinization, pelleted by centrifugation at 190 × $g$ for 5 min, and washed once with 5 mM HEPES (pH 7.4)-containing Hank's Balanced Salt Solution (HBSS). After centrifugation, the cells were resuspended in 8 mL of the HEPES-containing HBSS. The cell suspension was seeded in a 96-well culture plate (cell plate) at a volume of 90 µL (per well hereafter) and

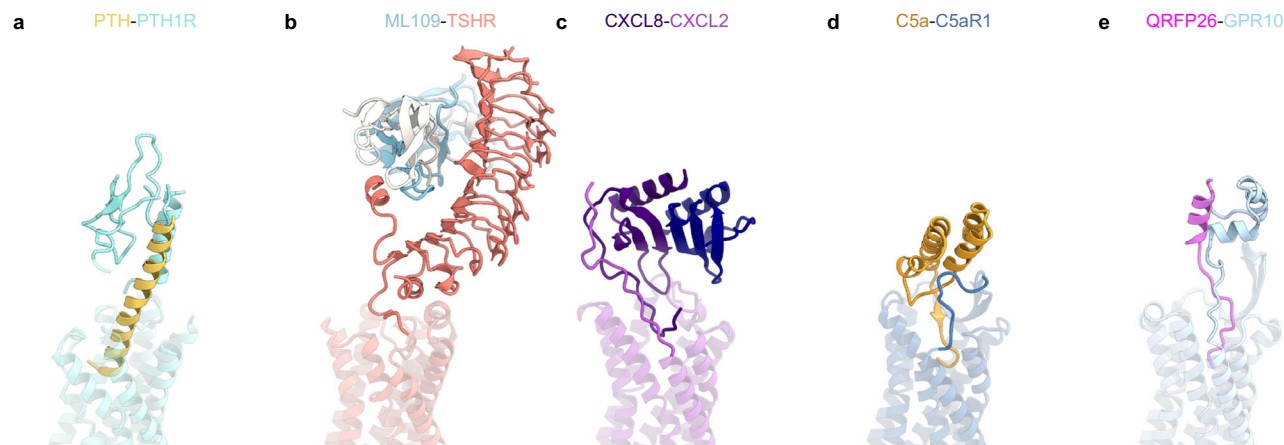

**Fig. 7 | Structural comparison of GPCRs with N-terminal regions. a–e** Structural comparison of the N-terminal region in PTH1R (PDB 7VVK(PTH-PTH1R–$G_s$ structure)) (**a**), TSHR (PDB 7XW5 (ML109–TSHR–$G_s$ structure)) (**b**), CXCR2 (PDB 8XWN (CXCL8–CXCR2 structure)) (**c**), C5aR (PDB 8HK5 (C5a–C5aR1–$G_i$ structure)) (**d**), and GPR103 (**e**).

incubated in a 37 °C incubator with 5% $CO_2$ for 30 min. The cells were mixed with 10 µL of titrated concentrations of 10× QRFP26 (Peptide institute) (diluted in HBSS containing 5 mM HEPES (pH 7.4) and 0.01% (w/v) fatty acid-free BSA (Serva)), and incubated for 60 min. After spinning the cell plates at 190 × $g$ for 2 min, 80 µL of conditioned media was transferred to an empty 96-well plate (conditioned media (CM) plate). Then, 80 µL of AP reaction solution (10 mM $p$-nitrophenylphosphate ($p$-NPP), 120 mM Tris-HCl (pH 9.5), 40 mM NaCl, and 10 mM $MgCl_2$) was dispensed into the cell plates and the CM plates. Absorbance at 405 nm of the plates was measured using a microplate reader (SpectraMax 340 PC384, Molecular Devices) before and after 1-h incubation at room temperature. Ligand-induced AP-TGFα release was calculated by subtracting spontaneous AP-TGFα release signal from ligand-induced AP-TGFα release signal. The AP-TGFα release signals were fitted to a four-parameter sigmoidal concentration–response curve, using the Prism 10 software (GraphPad Prism). For each replicate experiment, the $pEC_{50}$ parameter (negative logarithmic values of $EC_{50}$ values) of the individual GPR103 mutants were normalized to that of the wild-type GPR103 performed in parallel, and resulting $\Delta pEC_{50}$ values were used to denote ligand response activity of the mutants.

**Flow cytometry analysis**
Cell surface expression of GPR103 was measured as described previously[50]. HEK293A cells were seeded in a 6-well culture plate at a concentration of $2 \times 10^5$ cells per mL (2 mL per dish) and cultured for 1 day before transfection. The cells were transfected with N-terminally FLAG-tagged GPCR construct along with the AP-TGFα reporter by following the same procedure as described in the TGFα shedding assay section. After 1-day culture, the cells were collected by adding 200 µL of 0.53 mM EDTA-containing Dulbecco's PBS (D-PBS), followed by 200 µL of 5 mM HEPES (pH 7.4)-containing HBSS. The cell suspension was transferred to a 96-well V-bottom plate in duplicate and fluorescently labeled with an anti-Flag epitope (DYKDDDDK) tag monoclonal antibody (Clone 1E6, FujiFilm Wako Pure Chemicals) diluted in 2% goat serum and 2 mM EDTA-containing D-PBS (blocking buffer) and a goat anti-mouse IgG (H+L) secondary antibody conjugated with Alexa Fluor 488 (Thermo Fisher Scientific) diluted in the blocking buffer. After washing with D-PBS, the cells were resuspended in 200 µL of 2 mM EDTA-containing D-PBS and filtered through a 40-µm filter. The fluorescent intensity of single cells was quantified by a CytoFLEX flow cytometer (Beckman Coulter) equipped with a 488-nm laser. The flow cytometry data were analyzed with the FlowJo software (FlowJo). For each experiment, the

mean fluorescent intensity (MFI) value of mutants was normalized to that of the wild-type GPR103 performed in parallel. Values of mean fluorescence intensity from approximately 20,000 cells per sample were used for analysis.

**Expression and purification of the human GPR103 – $G_q$ complex**
The recombinant baculovirus was prepared using the Bac-to-Bac baculovirus expression system (Thermo Fisher Scientific). For expression, 0.8 L of HEK293S GnTI– (N-acetylglucosaminyl-transferase I−negative) cells (American Type Culture Collection, Cat. No. CRL-3022) at a density of $3 \times 10^6$ cells/mL were co-infected with baculovirus encoding GPR103, mini-$G_{sqi}$ trimer and $G_{\beta 1}$ at the ratio of 2:1:1. Twenty hours after infection, 10 mM of Sodium Butyrate was added, and the cells were incubated at 30 °C. After 48 h, the collected cells were resuspended and dounce-homogenized in 20 mM Tris-HCl, pH 8.0, 150 mM NaCl, 10% Glycerol, 4 µM QRFP26, 5.2 µg/mL aprotinin, 2.0 µg/mL leupeptin, and 100 µM PMSF. After homogenization, Apyrase was added to the lysis at a final concentration of 25 mU/mL, and the lysate was incubated at room temperature for 1 h. The crude membrane fraction was collected by ultracentrifugation at 180,000 × $g$ for 1 h and solubilized in buffer, containing 50 mM Tris-HCl, pH 8.0, 150 mM NaCl, 1.5% Lauryl Maltose Neopentyl Glycol (LMNG) (Anatrace), 0.15% cholesteryl hemisuccinate (CHS) (Merck), 10% glycerol, 5.2 µg/mL aprotinin, 2.0 µg/mL leupeptin, 100 µM PMSF, 25 mU/mL Apyrase, and 4 µM QRFP26 for 2 h at 4 °C. The supernatant was separated from the insoluble material by ultracentrifugation at 180,000 × $g$ for 30 min and incubated with 4 mL of Anti-DYKDDDDK G1 resin (Genscript) for 1 h at 4 °C. The resin was washed with 20 column volumes of buffer containing 20 mM Tris-HCl, pH 8.0, 500 mM NaCl, 10% Glycerol, 0.01% LMNG, 0.001% CHS, and 0.1 µM QRFP26. The complex was eluted in buffer containing 20 mM Tris-HCl, pH 8.0, 150 mM NaCl, 0.01% LMNG, 0.001% CHS, 10 µM QRFP26, and 0.2 mg/mL FLAG peptide. The eluate was incubated with the Nb35 and scFv16 at 4 °C. The complex was concentrated and purified by size-exclusion chromatography on a Superose 6 increase (GE) column in 20 mM Tris-HCl, pH 8.0, 150 mM NaCl, 0.01% LMNG, 0.001% CHS, and 0.1 µM QRFP26. QRFP26 was added to the peak fraction to the final concentration of 4 µM and concentrated up to 13.8 mg/ml.

**Sample vitrification and cryo-EM data acquisition**
The purified complex was applied onto a freshly glow-discharged Quantifoil holey carbon grid (R1.2/1.3, 300 mesh), and plunge-frozen in liquid ethane by using a Vitrobot Mark IV. Data collections were performed on a 300 kV Titan Krios G3i microscope (Thermo Fisher

Scientific) equipped with a BioQuantum K3 imaging filter and a K3 direct electron detector (Gatan).

First, 9555 movies were acquired with a calibrated pixel size of 0.83 Å pix$^{-1}$ and with a defocus range of −0.8 to −1.6 μm, using EPU. Each movie was acquired for 2.6 s and split into 48 frames, resulting in an accumulated exposure of about 49.6 electrons per Å$^2$ at the grid.

## Image processing

All acquired movies were dose-fractionated and subjected to beam-induced motion correction implemented in RELION 3.1[52]. The contrast transfer function (CTF) parameters were estimated using CTFFIND 4.0[53]. A total of 11,085,317 particles were extracted. The particles were subjected to 2D classifications, Ab-initio reconstruction, and several rounds of hetero refinement in cryoSPARC[32]. Next, the particles were re-extracted and exported to RELION 3.1[54], then subjected to 3D classification with a mask on the receptor. Then, the particles were subjected to Bayesian polishing in RELION[55]. The particle sets were exported to cryoSPARCv4.4.0, and subjected to CTF refinement, and Non-uniform refinement, yielding a map with a global nominal resolution of 3.19 Å, with the gold standard Fourier Shell Correlation (FSC = 0.143) criteria[56]. Moreover, the 3D model was refined with a mask on the receptor. As a result, the receptor has a higher resolution with a nominal resolution of 3.29 Å. The overall and receptor-focused maps were combined by phenix[57]. The processing strategy is described in Supplementary Fig. 3.

To investigate the flexibility of QRFP26 and ECD of GPR101, the batch of particles was further processed by 3D flexible refinement[32]. After an overall non-uniform refinement, the mesh was prepared using a micelle-removed density map. Then, following the 3D flex train job and the reconstruction job, the flexibility of the QRFP26 and the ECD of GPR103 were visualized by the flex generate job. The processing strategy is described in Supplementary Fig. 6.

## Model building and refinement

The density map was sharpened by phenix.auto_sharpen[58] and the quality of the density map was sufficient to build a model manually in COOT[59,60]. The model building was facilitated by the Alphafold-predicted structure and cryo-EM structure of OX$_2$R (PDB 7L1U (OxB-OX$_2$R-mini-G$_{sqi}$ structure))[12]. We manually fitted GPR103, the G$_q$ heterotrimer, and scFv16 into the map. We then manually readjusted the model using COOT and refined it using phenix.real_space_refine[57,61] (v.1.19) with the secondary-structure restraints using phenix secondary_structure_restraints. For the densities derived from 3D Flex, the rigid body refinement and all-atom refinement were performed on two representative frames from the default 40 frames using the model without 3D flex.

## Reporting summary

Further information on research design is available in the Nature Portfolio Reporting Summary linked to this article.

## Data availability

The cryo-EM density map and atomic coordinates for the QRFP26-GPR103 complex have been deposited in the Electron Microscopy Data Bank and the PDB, under accession codes: EMD-60096 and PDB 8ZH8. The tilted and upright state generated by the 3D are available at the Zenodo data repository (https://doi.org/10.5281/zenodo.11408292). All other data are available from the corresponding authors upon request. Source data are provided with this paper.

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

## Acknowledgements

We thank K. Ogomori and C. Harada for technical assistance; Kayo Sato, Shigeko Nakano, and Ayumi Inoue (Tohoku University) for their assistance in the cell-based assays; Masataka Yanagawa (Tohoku University) for the setup of the flow cytometer. This work was supported by grants from the JSPS KAKENHI, grant numbers JP21H05037 (O.N.), JP22K19371 and JP22H02751 (W.S.), and JP21H04791, JP21H05113, and JP21H05037 (Asuka I.); the ONO Medical Research Foundation (W.S.); the Kao Foundation for Arts and Sciences (W.S.); the Takeda Science Foundation (W.S.); the Uehara Memorial Foundation (W.S.); the Lotte Foundation (W.S.); Japan Science and Technology Agency (JST), grant numbers JPMJFR215T and JPMJMS2023 (Asuka I.); the Japan Agency for Medical Research and Development (AMED), grant numbers JP22ama121038 and JP22zf0127007 (Asuka I.), JP233fa627001 (O.N.); 22ck0106533h0003 (O.N.); and the Platform Project for Supporting Drug Discovery and Life Science Research (Basis for Supporting Innovative Drug Discovery and Life Science Research (BINDS)) from AMED, grant numbers JP23ama121002 (support number 3272, O.N.) and JP23ama121012 (supporting number 6204, Asuka.I.).

## Author contributions

Aika I. performed the cryo-EM structural study. R.K. and Asuka I. performed and oversaw the mutagenesis study. H.A., F.S., and H.S.O. assisted with the cryo-EM data collection and the single-particle analysis. W.S. designed the experiments. The manuscript was mainly prepared by Aika I., and W.S., with assistance from O.N. Asuka I., W.S., and O.N. supervised the research.

## Competing interests

O.N. is a co-founder and scientific advisor for Curreio. All other authors declare no competing interests.
