## [Peer Review File · Nature Communications]

Structure and dynamics of the pyroglutamylated RF-amide peptide QRFP receptor GPR103Reviewers' comments:

Reviewer #1 (Remarks to the Author):

This manuscript by Iwama et al. presents the cryo-EM structure of the RF-amide peptide receptor GPR103 in complex with Gq and the neuropeptide QRFP, which exhibits classic structural features in ligand recognition and receptor activation for Class A peptide GPCRs. A unique feature is that the N-terminal region of the receptor forms a helical structure to interact with the N-terminal domain of the ligand. Based on the structural analysis, the authors discussed the mechanism for ligand binding and selectivity. Overall, this is a high-quality structural study on a peptide GPCR.

Major issues:

1. There is no any functional data to support the structural insights. While the authors reference earlier mutagenesis studies, validating their structure through additional experiments is crucial. If nearly all interactions between the ligand and the receptor were predicted by previous studies, the structural insights in this paper may not significantly advance the field.

2. The 'two-site' binding mechanism has been well established for many Class A peptide GPCRs including chemokine receptors and the C5a receptor. Furthermore, the roles of the receptor N-termini in peptide ligand engagement have been well studied by multiple biophysical methods. Comparison with those GPCRs is important for elucidating the peptide recognition mechanism specific to GPR103.

3. The Supplementary Figure 2 indicates that the map resolution of the receptor's extracellular region, including the extracellular loops, is around 5Å after local refinement. This resolution is insufficient for precise modeling of the side chains. Therefore, it is important for the authors to show electron density maps of the critical residues of GPR103 involved in peptide binding in the Supplementary data to supporting their structural insights.

Other minor issues:

4. There is a discrepancy between the ligand-interacting residues shown in the main Figure 3 and those shown in the Supplementary Figure 3. The authors need to determine the accurate interacting residues and show them in the figures.

Reviewer #2 (Remarks to the Author):

In this study, Iwama et al. elucidated the structure of the QRFPR in complex with its ligand 26RFa and Gαq using cryo-EM. In the introduction, the authors present the family of RF-amide peptides and describe the peptides 43RFa and 26RFa and the corresponding receptor GPR103/QRFPR in more detail. In line 73, it is written, that the QRFPR is a Gαq-coupled receptor, however Ma et al. describe that GPR103 dually couples to Gαq and Gαi/o proteins (doi: 10.1016/j.bbamcr.2021.119046). Can the authors comment on this? The authors describe the physiological function of GPR103 and give an overview on related peptides, such as cholecystokinin, orexin and RY-amide neuropeptide Y and knowledge on the interaction with their corresponding GPCR. In line 71/72, the authors mention that QRFPR was identified by bioinformatics approach and reverse pharmacology. However, in Ref. 5, the authors described that it was identification by RT-PCR and in situ hybridization. Please, clarify and revise! In line 84, Iwama et al. introduced a GPR103 antagonist. Here, some additional information on the antagonist have to be added.

In the results section, the authors first describe the overall structure and the strategy used in this study. A C-terminally truncated variant of GPR103 was used (line 107). Please, provide data on the activity of this variant. Can the authors also comment why they used an engineered mini-Gsqi protein instead of native Gq protein? In line 126-127 the authors write: “Two disulfide bonds are observed in the GPR103 structure (Fig. 1e)” These interactions are not visible in the structure of Figure 1e. Also, these findings have to be discussed more carefully or in context with functional data, as the structure only gives information on spatial vicinity but not on physiologically relevant interactions. Further, it is described that 26RFa “adopts an elongated conformation consistent with the NMR analysis of QRFPR alone” (line 133-134). In this NMR study, Thuau et al. observed a nascent helix between residues 6-15 of 26RFa. Is this really consistent with the findings from the cryo-EM structure? The authors then describe the GPCR–Gα interface and write that residues within TM6 and TM7 are involved in an electrostatic network with the C-terminus of α5h (line 143/144). Here, a more detailed description have to be added. Additionally, it is of utmost importance to confirm key interaction points with mutagenesis data to proof the structural data. Contacts relevant for the function have to be distinguished from solely nearby residues. In line 146-148 it is stated, that “the bulky hydrophobic residue F151ICL2 is captured within a hydrophobic pocket in the Gα subunit, as in many other GPCRs^{16,21}”. However, in reference 21, an G protein mimetic antibody is used as substitute for the G protein. Maybe the authors can give more examples/references whether this hydrophobic interaction is known from many GPCRs and for which (including reference). The authors further describe the peptide binding site within the TM domain. Here they mention previous studies with mutant peptides (167-168) without giving any reference, this has to be added. In Figure 3b/c, it is hard to identify the relevant interactions between the C terminus of the peptide and the receptor TM core, described in lines 162-176. The authors must improve the visualization of the most important interactions or make two subfigures out of it. The authors found a unique architecture of the N-terminal region. The sentence “This N-terminal HLH motif is conserved among GPR103 homologs (Supplementary Fig. 4c), and our homology searches have failed to

identify any similar sequences in other proteins, indicating unique feature of GPR103.” (211-214) is a bit contradictory. Is the HLH motif unique to GPR103 or to RF-amide receptors? In line 220-236, the authors feature the N-terminal ECD configuration. Here, more detailed information is necessary. In the last part of the results section, the structure is compared to related GPCRs in “complex with Gq12-14” (line 257). However, the structure of the Y1R was determined with Gi1 protein. Please also add the corresponding PDB ID in the legend of Figure 5. It is described that “in Y1R, the nitrogen and C932.57 are uniquely positioned to form a hydrogen bond (Fig. 5c), a feature not found in GPR103”. This statement must be confirmed by mutagenesis studies. In line 273-278, the structures of NPY receptors are described. Here, reference 34 only includes the structure of Y1R and the respective reference describing the structure of Y1 and Y2 (Tang T et al., 2022) has to be added here.

In the discussion, the authors further analyze their findings. A more detailed analysis of the interaction between N-terminal HLH structure of GPR103 and N-terminal side of QRFP would be worth to gain new insights into this unusual interaction site. Also, it would be interesting if a N-terminal fragment of the peptide alone would be able to bind to the receptor without activating it, do the authors have any data on this? The authors describe “significant conformational changes” (line 326) within the ligand binding pocket, which are described between 1.4-1.6 Å. Is the resolution of the structure high enough to predict these changes?

Thus, all relevant contacts that have been found have to be confirmed by mutagenesis studies. Receptor mutants have to be cloned replacing the respective residue by e. g. alanine, receptor expression and activation of this receptor variant has to be compared to wild type receptor. This is state-of-the art to prove relevant contacts and distinguish functionally relevant from irrelevant ones. Structural information without functional data are meaningless. Furthermore, several references are missing and wording has to be more specific in some parts.

Minor comments:

Consistent spelling of RFamide/RF-amide/RF amide

general: better use QRFP26/43 or 26RFa/43RFa as QRFP is used for both peptides

37: GPCRs

91: Add blank space

105: 26RFa

106: 43RFa

133: conformation

149: R32G.hns1.03

187: better use names of receptors instead of GPR as ligands are known

246: including

335: In reference 38 a W6.48Q mutant is used, which doesn't really fit here

The revised parts in the text are highlighted in yellow. The major changes are as follows:

- We re-analyzed the cryo-EM data using cryoSPARC and improved the resolution of the extracellular domain. As a result, we corrected our model, in which the residues 5 to 13 of QRFP26 adopt an α -helix (Fig. 1e).
- We performed thorough mutant experiments (Supplementary Fig. 1a-d and Supplementary Table 1). The expression levels of the mutants were accurately quantified by fluorescence-activated cell sorting (FACS). As most of the mutants largely reduced their expression levels, we compared their activities with that of the wild type, whose expression levels were reduced to the same extent by decreasing the amount of used plasmid in transfection. This comparison allowed us to compare and discuss the mutational effects more accurately.
- We described a more detailed structural description of the ECD containing N-terminal HLH, which was validated by mutant experiments. Thus, we have made significant changes in the description of the relevant parts. Moreover, we compared the ECD with other receptors in Fig. 7 and discussed.
- Although G-protein interactions are one of the concepts of this paper, the main topic is the mechanism of peptide ligand recognition. Moreover, as reviewer 2 pointed out, we used the highly engineered G-protein for structural analysis, while it is commonly used in the structural analysis of GPCR-G_q complexes. Therefore, we have toned down the G-protein interactions, minimized references for it in the text, and depicted the interaction in Supplementary Fig. 4. We performed thorough mutant analysis, more focusing on the ligand binding site.
- To address the requirements of reviewer 2, we performed several additional experiments, including purification of the N-terminal expression of QRFP26 and evaluation of G_i/G_o activity.
- Finally, we correctly cited several essential papers that were missed in the first draft.

Reviewers' comments:

Reviewer #1 (Remarks to the Author):

This manuscript by Iwama etc. presents the cryo-EM structure of the RF-amide peptide receptor GPR103 in complex with Gq and the neuropeptide QRFP, which exhibits classic structural features in ligand recognition and receptor activation for Class A peptide GPCRs. A unique feature is that the N-terminal region of the receptor forms a helical structure to interact with the N-terminal domain of the ligand. Based on the structural analysis, the authors discussed the mechanism for ligand binding and selectivity. Overall, this is a high-quality structural study on a peptide GPCR.

Thank you very much for your evaluation of our paper. We believe that our paper was now much improved according to your suggestions.

Major issues:

1. There is no any functional data to support the structural insights. While the authors reference earlier mutagenesis studies, validating their structure through additional experiments is crucial. If nearly all interactions between the ligand and the receptor were predicted by previous studies, the structural insights in this paper may not significantly advance the field.

According to the suggestion, we made a total of 33 mutants of amino acid residues involved in peptide binding and evaluated their G_q activities by a TGF α shedding assay (Supplementary Fig. 1a-d, Supplementary Table 1). The expression levels of the mutants were accurately quantified by FACS. Mutants with significantly reduced expression were compared to the wild type, in which the plasmid amount was lowered to achieve the similar expression level.

2. The 'two-site' binding mechanism has been well established for many Class A peptide GPCRs including chemokine receptors and the C5a receptor. Furthermore, the roles of the receptor N-termini in peptide ligand engagement have been well studied by multiple biophysical methods. Comparison with those GPCRs is important for elucidating the peptide recognition mechanism specific to GPR103.

As the reviewer pointed out, the two site binding mechanism has been proposed in other class A GPCRs. According to the suggestion, we added the structural comparison with C5aR, CXCR2, and thyroid stimulating hormone receptor (TSHR) (Fig. 7) (lines 355 to 374). TSHR is a GPCR

with a large leucine-rich repeat domain at the N-terminus, but the transmembrane region is categorized as a class A GPCR, and thus we included the structural comparison.

3. The Supplementary Figure 2 indicates that the map resolution of the receptor's extracellular region, including the extracellular loops, is around 5Å after local refinement. This resolution is insufficient for precise modeling of the side chains. Therefore, it is important for the authors to show electron density maps of the critical residues of GPR103 involved in peptide binding in the Supplementary data to supporting their structural insights.

Thank you for your suggestion. We added the density map of the residues involved in peptide binding in Supplementary Fig. 3.

Other minor issues:

4. There is a discrepancy between the ligand-interacting residues shown in the main Figure 3 and those shown in the Supplementary Figure 3. The authors need to determine the accurate interacting residues and show them in the figures.

According to the suggestion, we reviewed the interacting residues and corrected the discrepancy between Fig. 3 (now Fig.2) and Supplementary Fig. 5. Moreover, we corrected the Fig. 2 more readable.

Reviewer #2 (Remarks to the Author):

In this study, Iwama et al. elucidated the structure of the QRFPR in complex with its ligand 26RFa and Gaq using cryo-EM. In the introduction, the authors present the family of RF-amide peptides and describe the peptides 43RFa and 26RFa and the corresponding receptor GPR103/QRFPR in more detail. In line 73, it is written, that the QRFPR is a Gaq-coupled receptor, however Ma et al. describe that GPR103 dually couples to Gaq and Gai/o proteins (doi: 10.1016/j.bbamcr.2021.119046). Can the authors comment on this?

We appreciate your thoughtful comments. We first performed a NanoBiT dissociation assay to confirm the activation of G_q and G_{i/o} by GPR103. The results showed activation of G_o by GPR103,

but not G_{i1} (Fig. L1a). Since the G_o activation is more sensitive than the G_i activation in this assay, this result indicates that GPR103 has the potential to activate $G_{i/o}$. Complicatedly, we could not measure the G_q activation in this assay (Fig. L1b). Because G_q -LgBiT is poorly expressed and more sensitive to endogenous G proteins than $G_{i/o}$, the G_q signal measured for other receptors is not as distinct as that of $G_{i/o}$. GPR103 has low membrane expression, which is likely to have a more pronounced effect. Overall, the NanoBiT G-protein dissociation assay does not discriminate which G-proteins are activated by GPR103. This paper focuses on G_q and the data is not clear enough to be included in the manuscript.

Thus, the G_q signal of GPR103 was measured using a TGF α shedding assay, and mutant analysis was performed with this assay (Fig 2,3 and Supplementary Fig. 1a-d, and Supplementary Table 1) This method avoided the problems described above as it uses endogenous G_q and allowed the measurement of G_q activation (Supplementary Fig. 1a-d).

Fig. L1 NanoBiT G-protein dissociation assay

The authors describe the physiological function of GPR103 and give an overview on related peptides, such as cholecystinin, orexin and RY-amide neuropeptide Y and knowledge on the interaction with their corresponding GPCR. In line 71/72, the authors mention that QRFP was identified by bioinformatics approach and reverse pharmacology. However, in Ref. 5, the authors described that it was identification by RT-PCR and in situ hybridization. Please, clarify and revise! In line 84,

We greatly appreciate your critical comment. There are three papers of the QRFP identification with different discovery methods. Thus, we removed the sentence “QRFP was identified by bioinformatics approach and reverse pharmacology” and simply wrote as “QRFP is a 43-amino

acid RF-amide peptide with a pyroglutamylated N-terminus (namely QRFP43) and demonstrates specific activity for GPR103 with orexigenic activity” with the citation of the papers (lines 72 to 74).

Iwama et al. introduced a GPR103 antagonist. Here, some additional information on the antagonist have to be added.

According to the suggestion, we added information about the antagonist as “Thus, GPR103 antagonists are expected to be useful in the prevention and treatment of various metabolic disorders such as bulimia, vasospasm, obesity, diabetes, endocrine disorders, hypercholesterolemia, hyperlipidemia, gout, and fatty liver. They may also be useful as therapeutic agents for cardiovascular and renal diseases, including atherosclerosis and heart failure. Several nonpeptidic GPR103 antagonists (e.g. pyrrolo[2,3-c]pyridine) have been discovered from peptidomimetics and high-throughput screening, and have demonstrated anorexic activity in mice” (lines 84 to 91).

In the results section, the authors first describe the overall structure and the strategy used in this study. A C-terminally truncated variant of GPR103 was used (line 107). Please, provide data on the activity of this variant.

This truncation did not affect the G_q activation in the TGF α shedding assay (Supplementary Fig. 1b-d and Supplementary Table 1). Truncation of the flexible C-terminal region is often favorable for structural biology because it typically improves the properties of the protein.

Can the authors also comment why they used an engineered mini-G_{sqi} protein instead of native G_q protein?

The reason is that wild-type G_q is poorly expressed, unstable and not suitable for structural analysis of complexes with GPCRs. Mini-G_{sqi} is a template of mini-G_s with seven heat-stabilizing mutations, and its C-terminal and N-terminal regions were replaced with those of G_q and G_i, respectively. In 2020, it was reported that the structure of G_q-coupled receptors can be efficiently determined using the mini-G_{sqi}, and since then, numerous G_q-coupled receptor structures have been reported using the same strategy, including orexin receptors, CCKR, the

type 2 bradykinin receptor, and others. Thus, the use of mini-G_{sqi} is a general strategy for the structure determination of Gq-coupled receptors.

In line 126-127 the authors write: “Two disulfide bonds are observed in the GPR103 structure (Fig. 1e)” These interactions are not visible in the structure of Figure 1e. Also, these findings have to be discussed more carefully or in context with functional data, as the structure only gives information on spatial vicinity but not on physiologically relevant interactions.

We added the disulfide bonds in Fig. 1e. We mutated the cysteines and added the description as “Two disulfide bonds are represented by stick models. One is the highly conserved disulfide bond between C118^{3.25} and C201^{ECL2}, and the other is atypical disulfide bond between C285^{6.47} and C327^{7.48}. The C118^{3.25}A and C201^{ECL2}A mutations abolished the QRFP26 potency (Supplementary Fig. 1a–d). By contrast, the C285^{6.47}A and C327^{7.48}A mutations did not alter the potency, indicating less importance for receptor function” (lines 402 to 407 in the figure legend of Fig. 1e).

Further, it is described that 26RFa “adopts an elongated conformation consistent with the NMR analysis of QRFP alone” (line133-134). In this NMR study, Thuau et al. observed a nascent helix between residues 6-15 of 26RFa. Is this really consistent with the findings from the cryo-EM structure?

We re-analyzed the cryo-EM data using cryoSPARC and improved the resolution of the overall structure to 3.19 Å and receptor to 3.29Å. As a result, we corrected our model, in which the residues 5 to 13 of QRFP26 adopt an α-helix (Fig.1e and Supplementary Fig. 3), which is now consistent with the NMR study (Thuan *et al.*).

The authors then describe the GPCR–Gα interface and write that residues within TM6 and TM7 are involved in an electrostatic network with the C-terminus of α5h (line 143/144). Here, a more detailed description have to be added. Additionally, it is of utmost importance to confirm key interaction points with mutagenesis data to proof the structural data. Contacts relevant for the function have to be distinguished from solely nearby residues.

Although G-protein interactions are one of the concepts of this paper, the main topic is the mechanism of peptide ligand recognition. For this reason, we have toned down the G-protein

interactions, minimized references for it in the main text, and depicted the interaction in Supplementary Fig. 4.

In line 146-148 it is stated, that “the bulky hydrophobic residue F151ICL2 is captured within a hydrophobic pocket in the G α subunit, as in many other GPCRs^{16,21}”. However, in reference 21, an G protein mimetic antibody is used as substitute for the G protein. Maybe the authors can give more examples/references whether this hydrophobic interaction is known from many GPCRs and for which (including reference).

Thank you for your critical points. We cited the wrong reference in the previous manuscript. This hydrophobic interaction is known from many GPCRs, and thus we added the comparison of the ICL2-G α interface with the representative GPCR-G-protein complex (Supplementary Fig. 4c-i).

The authors further describe the peptide binding site within the TM domain.

We revised the description about the peptide binding site in the TM domain with the results of the mutational studies (lines 143 to 176).

Here they mention previous studies with mutant peptides (167-168) without giving any reference, this has to be added.

We added the references (line 156).

In Figure 3b/c, it is hard to identify the relevant interactions between the C terminus of the peptide and the receptor TM core, described in lines 162-176. The authors must improve the visualization of the most important interactions or make two subfigures out of it.

According to the suggestion, we have fully revised Fig. 3 (now Fig. 2).

The authors found a unique architecture of the N-terminal region. The sentence “This N-terminal HLH motif is conserved among GPR103 homologs (Supplementary Fig. 4c), and our homology searches have failed to identify any similar sequences in other proteins, indicating unique feature of GPR103.” (211-214) is a bit contradictory. Is the HLH motif unique to GPR103 or to RF-amide receptors?

In other RFamide receptors, the N-terminal HLH of GPR103 is not conserved at all. In this part, GPR103 homologs refer to GPR103 from various species, not to other RF amide receptors. We intended to argue that the N-terminal HLH is evolutionarily conserved. Thus, we revised the sentence as “This N-terminal HLH motif is evolutionarily conserved among GPR103 homologs from zebrafish to human.” (lines 220 and 221).

In line 220-236, the authors feature the N-terminal ECD configuration. Here, more detailed information is necessary.

We added the further description about the HLH in Fig. 3 and text (lines 195 to 201).

In the last part of the results section, the structure is compared to related GPCRs in “complex with Gq12-14” (line 257). However, the structure of the Y1R was determined with Gi1 protein. According to the suggestion, we corrected as “the endogenous ligand-bound structures of Y1R, CCK1R, and OX2R” (lines 274 and 275).

Please also add the corresponding PDB ID in the legend of Figure 5.

According to the suggestion, we added the IDs.

It is described that “in Y1R, the nitrogen and C932.57 are uniquely positioned to form a hydrogen bond (Fig. 5c), a feature not found in GPR103”. This statement must be confirmed by mutagenesis studies.

We performed the mutational analysis of C98^{2.57}. The C98^{2.57}A mutation only reduced the QRFP26 potency by 3-fold. We revised the sentence as “It should be noted that C^{2.57} is completely conserved and forms a hydrogen bond with the C-terminal nitrogen of NPY in Y₁R, while the hydrogen bond is doubtful in GPR103 due to bond angle issue. However, the C98^{2.57}A mutation reduced the QRFP potency by 3-fold, suggesting the potential involvement in QRFP26 binding (Fig. 5e).” (lines 289 to 293).

In line 273-278, the structures of NPY receptors are described. Here, reference 34 only includes the structure of Y1R and the respective reference describing the structure of Y1 and Y2 (Tang T et al., 2022) has to be added here.

We added the citation of Tang T et al, 2022 and Kang, H. et al, 2023.

In the discussion, the authors further analyze their findings. A more detailed analysis of the interaction between N-terminal HLH structure of GPR103 and N-terminal side of QRFP would be worth to gain new insights into this unusual interaction site. Also, it would be interesting if a N-terminal fragment of the peptide alone would be able to bind to the receptor without activating it, do the authors have any data on this?

As requested by the reviewer, we expressed and purified the N-terminal region (1-44) with a His tag at the C-terminus in *E. coli*. Although we tried several conditions, we could not obtain a band of the target protein other than the contaminants by the deadline (Fig. L2). As the reviewer pointed out, we are also interested in whether N-terminal HLH can function alone, as the ECD of PTH1R. However, the N-terminal HLH is so small that it would not be able to maintain its current conformation without hydrophobic interactions with ECL2. Moreover, the amino acids comprising the N-terminal HLH are mainly hydrophobic (16/44) and thus the sample may be highly hydrophobic and difficult in solubilization without the aid of the other part of GPR103.

Fig. L2 Purification of the N-terminal region-His6.

SDS-PAGE result of eluate from Ni-NTA resin

The authors describe “significant conformational changes” (line 326) within the ligand binding pocket, which are described between 1.4-1.6 Å. Is the resolution of the structure high enough to predict these changes?

This paragraph proposes a mechanism for receptor activation by comparing the predicted structure in the inactive state. Thus, as you pointed out, the validity of the movement of the structural change is speculative, and of course the word "significant" is not appropriate. We removed the word "significant". Moreover, we performed mutant experiments on F282, W286, and F289, whose importance is discussed in this chapter. The mutants F282A, W286A, and F289A almost abolished the potency of QRFP26, supporting our proposed importance for receptor activation (Supplementary Fig. 9f).

Thus, all relevant contacts that have been found have to be confirmed by mutagenesis studies. Receptor mutants have to be cloned replacing the respective residue by e. g. alanine, receptor expression and activation of this receptor variant has to be compared to wild type receptor. This is state-of-the art to prove relevant contacts and distinguish functionally relevant from irrelevant ones. Structural information without functional data are meaningless. Furthermore, several references are missing and wording has to be more specific in some parts.

According to the suggestion, we made 33 mutants, in total, of amino acid residues involved in peptide binding and evaluated their G_q activities by a TGF α shedding assay (Supplementary Fig. 1a-d, Supplementary Table 1). The expression levels of the mutants were accurately quantified by FACS. Mutants with significantly reduced expression were compared to the wild type, in which the plasmid amount was lowered to achieve the similar expression level. We revised the manuscript describing the interaction between QRFP26 and GPR103 to integrate the mutagenesis result. Moreover, we correctly cited several essential papers that were missed in the first draft.

Minor comments:

✓ Consistent spelling of RFamide/RF-amide/RF amide

general: better use QRFP26/43 or 26RFa/43RFa as QRFP is used for both peptides

We unified the words as "RF-amide" and "QRFP26/43".

✓37: GPCRs

✓91: Add blank space

We corrected the words.

✓105: 26RFa

✓106: 43RFa

We corrected the words as QRFP26 and QRFP43.

✓133: conformation

149: R32G.hns1.03

We corrected the words.

187: better use names of receptors instead of GPR as ligands are known

We used both names of receptors and GPRs.

✓246: including

We corrected the word.

335: In reference 38 a W6.48Q mutant is used, which doesn't really fit here

We removed the citation.

REVIEWERS' COMMENTS

Reviewer #1 (Remarks to the Author):

The authors have addressed all of my concerns. I appreciate the additional functional data and the new cryo-EM map provided by the authors.

Reviewer #2 (Remarks to the Author):

All our comments have been considered, all questions we raised have been answered.

There are only a few typos that should be changed:

- Figure 5e: E45.32 & D/E45.32  E45.52 & D/E45.52?
- Supplementary Figure 1b: 2xWT(1:1)  WT(1:50)?
- Supplementary Figure 2: Sample purification